# Generative Models for Effective ML on Private, Decentralized Datasets

**Sean Augenstein**
Google Inc.
saugenst@google.com

**H. Brendan McMahan**
Google Inc.
mcmahan@google.com

**Daniel Ramage**
Google Inc.
dramage@google.com

**Swaroop Ramaswamy**
Google Inc.
swaroopram@google.com

**Peter Kairouz**
Google Inc.
kairouz@google.com

**Mingqing Chen**
Google Inc.
mingqing@google.com

**Rajiv Mathews**
Google Inc.
mathews@google.com

**Blaise Aguera y Arcas**
Google Inc.
blaisea@google.com

## Abstract

To improve real-world applications of machine learning, experienced modelers develop intuition about their datasets, their models, and how the two interact. Manual inspection of raw data—of representative samples, of outliers, of misclassifications—is an essential tool in a) identifying and fixing problems in the data, b) generating new modeling hypotheses, and c) assigning or refining human-provided labels. However, manual data inspection is problematic for privacy-sensitive datasets, such as those representing the behavior of real-world individuals. Furthermore, manual data inspection is impossible in the increasingly important setting of federated learning, where raw examples are stored at the edge and the modeler may only access aggregated outputs such as metrics or model parameters. This paper demonstrates that generative models—trained using federated methods and with formal differential privacy guarantees—can be used effectively to debug many commonly occurring data issues even when the data cannot be directly inspected. We explore these methods in applications to text with differentially private federated RNNs and to images using a novel algorithm for differentially private federated GANs.

## 1 Introduction

Real-world systems increasingly depend on machine learning (ML) to make decisions, detect anomalies, and power products. Applications ranging from fraud detection to mobile phone keyboards use models trained on data that may be privacy sensitive. The data may contain financial, medical, or behavioral information about individuals. Institutions responsible for these applications of ML must balance data stewardship obligations—including minimizing the risks of data loss, theft, or abuse—with the practical needs of the "modeler" whose job is to develop and improve the machine learned models.

Modelers working with privacy-sensitive data face significant challenges in model development and debugging. Often, a modeler's first step would be to inspect individual examples in order to discover bugs, generate hypotheses, improve labeling, or similar. But direct inspection of data may be audited or disallowed by policy in some settings. In other settings—including federated learning (FL) (McMahan & Ramage, 2017;

McMahan et al., 2017), which is the motivation and focus of this work—the data *cannot* be inspected. In FL, raw data remains distributed across a fleet of devices, such as mobile phones, while an orchestrating server coordinates training of a shared global model. Only the final model parameters and statistics are gathered and made available to the modeler. [1]

How can a modeler effectively debug when training data is privacy sensitive or decentralized? This paper demonstrates that the novel application of auxiliary models, namely privacy-preserving generative models, can stand in for direct data examination during the process of debugging data errors during inference or training. By combining ideas from deep generative models, FL, and user-level differential privacy (DP), we show how some needs traditionally met with data inspection can instead be met by generating synthetic examples from a privacy-preserving federated generative model. These examples could be representative of all or a subset of the non-inspectable data, while at the same time preserving the privacy of individuals. Our contributions include:

- Identifying key challenges in implementing end-to-end workflows with non-inspectable data, e.g., for debugging a 'primary' ML model used in a mobile application.
- Proposing a methodology that allows (sufficiently powerful) 'auxiliary' generative models to resolve these challenges.
- Demonstrating how privacy preserving federated generative models—RNNs for text and GANs for images—can be trained to high enough fidelity to discover introduced data errors matching those encountered in real world scenarios. This requires a novel adaption of generative adversarial networks (GANs) to the federated setting with user-level DP guarantees.

## 2 CHALLENGES OF ML ON NON-INSPECTABLE DATA

A key aspect of deep learning is the utilization of large datasets. The behavior of a deep network is tightly coupled to the data it trains on. As Sculley et al. (2015) puts it, "ML is required in exactly those cases when the desired behavior cannot be effectively expressed in software logic without dependency on external data." The centrality of data to ML performance is recognized in the attention paid to data analysis, curation, and debugging. Textbooks devote chapters to methodologies for "Determining Whether to Gather More Data" (Goodfellow et al., 2016). The literature conveys practical lessons learned on ML system anti-patterns that increase the chance of data processing bugs (Sculley et al., 2015). In general, an assumption of unfettered access to training or inference data is made (Riley, 2016; Zinkevich, 2017; Chakarov et al., 2016). What to do if direct access is precluded, e.g., if working with private and decentralized data, is largely unaddressed.

Below, we describe six common tasks (**T1**–**T6**, summarized in Table 1) where a modeler would typically use direct data access. Our choice of these tasks is validated by Humbatova et al. (2019), a recent survey providing a taxonomy of faults in deep learning systems. Two of the largest classes of faults are in 'Preprocessing of Training Data' and 'Training Data Quality'; **T1**–**T6** are manners of addressing these faults.

**Sanity checking and model debugging (T1–T4)**  Experienced modelers will often inspect some random examples and observe their properties before training a model (**T1**); or if this step is skipped, it is often a first step in debugging. Are the size, data types, and value ranges as expected? For text data, are words or word pieces being properly separated and tokenized? A modeler is often working to identify mistakes in upstream processing, affecting all or most data, and readily apparent by looking at one or a few input samples.

When a model is misbehaving, looking at a particular subset of the input data is natural. For example, in a classification task, a modeler might inspect misclassified examples to look for issues in the features or

---

[1]Despite these limitations, FL systems (Bonawitz et al., 2019) have proven effective for training and deploying real-world models, for example see Hard et al. (2018); Yang et al. (2018); Chen et al. (2019).

Table 1: ML modeler tasks typically accomplished via data inspection. In Section 2.1 we observe that selection criteria can be applied programmatically to train generative models able to address these tasks.

| | Task | Selection criteria for data to inspect |
|---|---|---|
| **T1** | Sanity checking data | Random training examples |
| **T2** | Debugging mistakes | Misclassified examples (by the primary classifier) |
| **T3** | Debugging unknown labels/classes, e.g. out-of-vocabulary words | Examples of the unknown labels/classes |
| **T4** | Debugging poor performance on certain classes/slices/users | Examples from the low-accuracy classes/slices/users |
| **T5** | Human labeling of examples | Unlabeled examples from the training distribution |
| **T6** | Detecting bias in the training data | Examples with high density in the serving distribution but low density in the training distribution. |

labels (**T2**). For tasks where the full set of possible labels is too large, e.g. a language model with a fixed vocabulary, the modeler might examine a sample of out-of-vocabulary words (**T3**). For production ML applications, it is often important to monitor not only overall accuracy, but accuracy over finer grained slices of the data — for example, by country, by class label, by time of day, etc.[2] If low accuracy is observed on such a slice, it is natural to examine examples selected from that segment of the data (**T4**). For example, if training on data which can be grouped by user, it is natural (but potentially problematic from a privacy perspective) to look at data from users with low overall accuracy.

**Data labeling (T5)** Supervised learning problems require labeled data. Often in FL, labels can be inferred naturally from context on device—for example, the next-word predictor in a keyboard app (e.g., as in Hard et al. (2018)) is a problem where the input is a sequence of words and the label is the subsequent word. But some supervised learning problems are better suited to human raters inspecting and manually assigning class labels (e.g., visual object recognition datasets as in Deng et al. (2009)). Because FL systems do not allow users' data to be sent to the cloud for labeling, it would be desirable to synthesize realistic, representative examples from the decentralized data that could be labeled by human raters. Doing so would require a private mechanism of generating high-fidelity training examples. This would expand the set of ML problems that can be afforded FL's privacy advantages.

**Detecting bias in training data (T6)** Another common use of human raters in supervised learning is to assign labels to non-privacy-sensitive training examples curated from a public dataset or collected via donation. These examples might be the only ones a classifier is trained on, even if a much larger unlabeled dataset exists (e.g. on phones in FL). The distributional difference between the datasets is a source of bias. For example, a labeled training dataset for a face detector might not be representative of the diversity of racial/ethnic/gender/age classes that the face detector sees through a phone's camera (i.e., at serving time). These classes can be unrelated to the model's task but serve as a confounding factor in prediction if the model has erroneously fit an assumption about them during training (e.g., a smiling face detector that judges smiling vs. non-smiling based on gender or gender expression (Denton et al., 2019)). When entire groups are un- or underrepresented in training, the result can be biased predictions that induce quite insidious social effects (Angwin et al., 2016). Modelers' goals of building unbiased classifiers could be met by inspecting examples of classes missing at training but present at prediction time. These examples—even at low fidelity—could indicate whether a model is fair and point to where additional training data collection is required.

---

[2]E.g., McMahan et al. (2013, Sec. 5.2) notes 100s of slices of relevance for ad-click-through-rate prediction.

## 2.1 USING GENERATIVE MODELS INSTEAD OF DATA INSPECTION

The general methodology for using generative models in place of data inspection is as follows: in a situation where the modeler would otherwise inspect examples based on a particular criteria (as shown in Table 1), this criteria is expressed as a programmatic data selection procedure used to construct a training dataset for a generative model. In the case of FL specifically, this might involve both selecting only a subset of devices to train the model, as well as filtering the local dataset held on each device.

We conduct experiments for three such tasks in this paper. Section 5 considers **T3**, and shows how the methodology proposed here can be extended when the primary model is itself a generative model. We conduct experiments for **T2** and **T4** in Section 6, where both approaches provide alternative ways of discovering the same bug; results for the **T4** approach are presented in the main body, with **T2** results appearing in the Appendix. We chose these scenarios as being representative of a large class of data-related ML bugs, based on real-world experience in both industry and academic settings. Independently, Humbatova et al. (2019) identifies 'text segmentation' and 'pixel encoding' as being very common sources of bugs. By showing that privacy-preserving federated generative models can debug these scenarios, we demonstrate the applicability of the approach. One of the reasons we select these scenarios is that high-fidelity generation is not necessary to detect these bugs, and pushing the state-of-the-art in generative models is not the aim of this work.

In contrast, for some classes of bugs, as well for generating examples used for labeling and training (**T5**), high-fidelity private generative models are necessary. We hope that this work will serve as a call-to-action for the generative modeling community, as our work strongly suggests that these models have a fundamental role to play in enabling the widespread use of privacy-preserving ML workflows.

Task **T6** requires a more subtle approach in defining the selection criteria. We briefly describe possible approaches in the spirit of providing a more complete description of the uses of generative models in private ML workflows, but leave details and experiments as promising directions for future work. Suppose we have a public, centralized distribution $\mathcal{P}_{\text{public}}$ as well as a private, decentralized distribution $\mathcal{P}_{\text{private}}$. Informally, we might wish to generate examples according to $x \sim \frac{\mathcal{P}_{\text{private}}(x)}{\mathcal{P}_{\text{public}}(x)}$ or perhaps $x \sim \mathcal{P}_{\text{private}}\left(x \mid \mathcal{P}_{\text{public}}(x) \le \epsilon\right)$. If we have a family of generative models for the domain of $x$ that explicitly provide a likelihood (e.g., the language models we consider in Section 5), then we can first train such a generative model on $\mathcal{P}_{\text{public}}$, and use this to filter or re-weight the decentralized samples drawn from $\mathcal{P}_{\text{private}}$ to train a second generative model which will be used to generate samples representative of $\mathcal{P}_{\text{private}}$ but not $\mathcal{P}_{\text{public}}$. In the case of architectures that do not provide an explicit likelihood (e.g., GANs), alternative techiques are needed, for example a secondary discriminator used to differentiate samples from $\mathcal{P}_{\text{private}}$ from those from $\mathcal{P}_{\text{public}}$.

Of course, if any of the generative models used here simply reproduce the private examples of users, nothing has been gained. Hence, we next turn to training these models with strong privacy guarantees.

## 3 DIFFERENTIALLY PRIVATE FEDERATED GENERATIVE MODELS

To implement the approach of Section 2.1, we propose combining three technologies: generative models, federated learning (FL), and differential privacy (DP). Deep generative models can synthesize novel examples, FL can train and evaluate against distributed data, and FL and DP both afford user privacy protections.

Critically, none of the challenges presented require the inspection of any particular user's data. As with ML, the goal is to discover something broadly meaningful. Thus, it is natural to consider the use of suitable synthetic examples in lieu of real user data. For this, we can leverage generative models based on deep neural networks (or 'deep generative models'). In contrast to discriminative models, generative models learn a joint distribution $p(x, y)$ and can be applied to data *synthesis*. Such neural networks can either approximate a likelihood function or serve as a mechanism for drawing samples (i.e., an implicit distribution). Deep generative

models of various forms (Kingma & Welling, 2013; Goodfellow et al., 2014; Kumar et al., 2019; Radford et al., 2019) have garnered a considerable amount of research interest, particularly for high dimensional spaces where explicit modeling is intractable. Application domains include text, audio, and imagery.

To work with decentralized data, we train these generative models via FL, ensuring raw user data never leaves the edge device. Instead, a randomly chosen subset of devices download the current model, and each locally computes a model update based on their own data. Ephemeral model updates are then sent back to the coordinating server, where they are aggregated and used to update the global model. This process repeats over many rounds until the model converges (for more information, see McMahan et al. (2017) and Bonawitz et al. (2019)). FL is borne of the mobile domain, where privacy is paramount and data is decentralized. Privacy for the mobile domain is a major motivator for this work, and we'll use FL extensively in this paper.

As with other ML models, deep generative models have a tendency to memorize unique training examples, leading to a valid concern that they may leak personal information. Differential privacy is a powerful tool for preventing such memorization. In particular, in this paper we'll emphasize the use of *user*-level DP, obtained in the FL context via a combination of per-user update clipping and post-aggregation Gaussian noising, following McMahan et al. (2018); Appendix A reviews user-level DP and FL. We assume this mechanism is implemented by the FL infrastructure, before the modeler has access to the trained models. This approach bounds the privacy loss to both the modeler and the devices that participate in FL rounds. In our experiments, we quantify the privacy obtained using $(\epsilon, \delta)$ upper bounds.

## 4  RELATED WORK

Prior work has looked at training GANs with DP in a centralized setting (Torkzadehmahani et al., 2019; Xie et al., 2018; Zhang et al., 2018; Frigerio et al., 2019; Beaulieu-Jones et al., 2018; Esteban et al., 2017). The focus is on how privacy protections like DP can be leveraged to quantifiably guarantee that a GAN will not synthesis examples too similar to real data. However, this work does not apply to training on decentralized data with FL, nor does it consider user-level DP guarantees, both of which are critical for our applications.

Other work studies generative models for decentralized data problems, but without privacy protections. An example is training GANs (Hardy et al., 2018) on decentralized data across multiple data centers. We consider this conceptually distant from the topic of this paper, where user-level privacy protection is paramount.

Two recent papers focus on what could be considered as specific instances of generative FL-based tasks, without connecting those instances to a much larger class of workflows on decentralized data in need of a general solution (i.e., all of **T1**-**T6** from Table 1). First, the FedGP algorithm of Triastcyn & Faltings (2019) is an approach to building a federated data synthesis engine based on GANs, to address a specific task resembling **T5**. Where we use DP as our means of measuring and protecting against data release (providing a worst-case bound on privacy loss), FedGP adopts a weaker, empirical measure of privacy (specifically, differential *average-case* privacy) which has not yet been broadly accepted by the privacy community. Second, recent language modeling work in Chen et al. (2019) uses a character-level RNN trained via FL to perform the **T3** task of generating common out-of-vocabulary (OOV) words (i.e., words that fall outside a finite vocabulary set). OOVs are interesting to the language modeler as they reflect natural temporal changes in word usage frequency (e.g., new slang terms, names in the news, etc.) that should be tracked. Their work does not apply DP. We adopt this OOV model for debugging purposes (and add DP) in experiments in Section 5.

## 5  AN APPLICATION TO DEBUGGING DURING TRAINING WITH RNNS

**DP Federated RNNs for Generating Natural Language Data**   Recurrent Neural Networks (RNNs) are a ubiquitous form of deep network, used to learn sequential content (e.g., language modeling). An interesting property of RNNs is that they embody both discriminative and generative behaviors in a single model. With

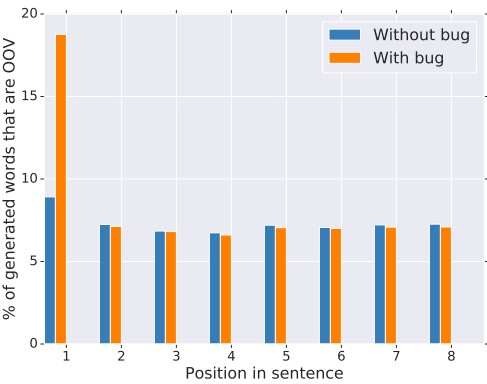

Figure 1: Percentage of samples generated from the word-LM that are OOV by position in the sentence, with and without bug.

Table 2: Top 10 generated OOV words by joint character probability (computed using Equation 1), with and without the bug. Number accompanying is joint probability. The model is trained with case-insensitive tokenization.

| *Without bug (prob $10^{-3}$)* | | *With 10% bug (prob $10^{-3}$)* | |
|---|---|---|---|
| regex | 5.50 | i have | 8.08 |
| jsfiddle | 3.90 | i am | 5.60 |
| xcode | 3.12 | regex | 4.45 |
| divs | 2.75 | you can | 3.71 |
| stackoverflow | 2.75 | if you | 2.81 |
| listview | 2.74 | this is | 2.77 |
| textbox | 2.73 | here is | 2.73 |
| foreach | 2.34 | jsfiddle | 2.70 |
| async | 2.27 | i want | 2.39 |
| iis | 2.21 | textbox | 2.28 |

regards to the former, an RNN learns a conditional distribution $p(x_{i+1}|x_i, \ldots, x_0; D)$. Given a sequence of tokens $x_0, \ldots, x_i$ and their successor $x_{i+1}$, the RNN reports the probability of $x_{i+1}$ conditioned on its predecessors (based on observing dataset $D$). Typically, $x_0$ is a special 'beginning-of-sequence' token.

But an RNN is also a generative model: the successor token probability distribution can be sampled from to generate a next token, and this process can be repeated to generate additional tokens in the sequence. The repeated application of the RNN provides a sample from the following joint distribution:

$$p(x_n, x_{n-1}, \ldots, x_2, x_1, x_0|D) = \prod_{i=0}^{n-1} p(x_{i+1}|x_i, \ldots, x_0; D) \cdot p(x_0|D) \tag{1}$$

In this way, RNN word language models (word-LMs) can generate sentences and RNN character language models (char-LMs) can generate words. Training an RNN under FL and DP can be performed via the DP-FedAvg algorithm of McMahan et al. (2018), restated in Appendix A, Algorithm 2.

**Experiment** Consider a mobile keyboard app that uses a word-LM to offer next-word predictions to users based on previous words. The app takes as input raw text, performs preprocessing (e.g. tokenization and normalization), and then feeds the processed list of words as input to the word-LM. The word-LM has a fixed vocabulary of words $V$. Any words in the input outside of this vocabulary are marked as 'out-of-vocabulary' (OOV). Such word-LMs have been trained on-device via FL (Hard et al., 2018). We demonstrate how we can detect a bug in the training pipeline for the word-LM using DP federated RNNs.

Suppose a bug is introduced in tokenization which incorrectly concatenates the first two tokens in some sentences into one token. E.g., 'Good day to you.' is tokenized into a list of words as ['Good day', 'to', ...], instead of ['Good', 'day', 'to', ...]. Such issues are not uncommon in production natural language processing (NLP) systems. Because these concatenated tokens don't match words in vocabulary $V$, they will be marked as OOVs. Under normal circumstances OOVs occur at a relatively consistent rate, so this bug will be noticed via a spike in OOV frequency, a metric an NLP modeler would typically track (see Appendix B). Were this the cloud, some of the OOV tokens could be inspected, and the erroneous concatenation realized. But here the dataset is private and decentralized, so inspection is precluded.

We simulate the scenario using a dataset derived from Stack Overflow questions and answers (hosted by TensorFlow Federated (Ingerman & Ostrowski, 2019)). It provides a realistic proxy for federated data, since we can associate all posts by an author as a particular user (see Appendix B.1 for details). We artificially introduce the bug in the dataset by concatenating the first two tokens in 10% of all sentences, across users.

Two federated generative models are used in complementary fashion for debugging. The first is the 'primary' model, the DP word-LM that is being trained via FL for use in the next word prediction pipeline. While the bug is inhibiting its training for production use, we can still harness its generative properties for insight into the nature of the bug. The second is an 'auxiliary' model for **T3**, a DP char-LM trained only on OOV words in the dataset; this model is taken from Chen et al. (2019), but here our goal is debugging, not learning new words. We assume we have access to both the word-LM and char-LM trained before the bug was introduced, as well as both models trained on buggy data. Continuous training of auxiliary models is beneficial in a production workflow, to contrast output of models trained at different times or on different selection criteria.

Both the DP word- and char-LMs are trained independently via simulated FL (following the approach of McMahan et al. (2018)) on the bug-augmented and bug-free Stack Overflow dataset, producing four trained models in total. These models are now used to synthesize useful and complementary information about the pipeline's input data. From Figure 1, we notice that with the bug the sentences generated by the word-LM have a larger than normal OOV rate for the first position. In Table 2, we see that with the bug most of the top OOV words generated from the char-LM have spaces in them, indicating a tokenization bug in the pipeline. This bug can then be fixed by the modeler. See Appendix B for further details and expanded results.

**Privacy**   The dataset contains 342,477 unique users. Each round, 5,000 random users are selected[3]. The models trained for 2,000 communication rounds with DP hyperparameters as in Appendix B.2, Table 5. As an example, this gives $\epsilon = 9.22$ with $\delta = 2.92 \times 10^{-6}$. A larger number of users would lead to a smaller $\epsilon$.

## 6   AN APPLICATION TO DEBUGGING DURING INFERENCE WITH GANS

**DP Federated GANs for Generating Image Data**   Generative Adversarial Networks (GANs) (Goodfellow et al., 2014) are a state-of-the-art form of deep generative model, with numerous recent successes in particular in the image domain (Isola et al., 2016; Zhu et al., 2017; Karras et al., 2018; 2019; Brock et al., 2019). GANs work by alternately training two networks. One is a generator which maps a random input vector in a low-dimensional latent space into a rich, high-dimensional generative output like an image. The other is a discriminator, which judges whether an input image is 'real' (originating from a dataset of actual images) or 'fake' (created by the generator). Each network tries to defeat the other; the generator's training objective is to create content that the discriminator is unable to discern from real content, and the discriminator's training objective is to improve its ability to discern real content from generated content.

We can take the GAN training framework and adapt it to FL and DP, analogous to RNN training under FL and DP in McMahan et al. (2018) via the DP-FedAvg algorithm. The difference here is that two sets of model parameters are updated, by alternating minimization of two loss functions. One key insight is that only the discriminator's training step involves the use of real user data (private and restricted to the user's device); the generator training step does not require real user data, and thus can be computed at the coordinating server via a traditional (non-federated) gradient update. A second insight is that the generator's loss is a function of the discriminator. As observed by earlier works involving DP GANs, if the discriminator is trained under DP and the generator is trained only via the discriminator, then the generator has the same level of privacy as the discriminator, via the post-processing property of DP (Dwork & Roth, 2014). No additional computational steps (e.g., clipping, noising) are necessary at the application of the generator gradient update.

---

[3]These experiments used sampling with replacement instead of without replacement due to a technical limitation of the simulation framework; however, this difference should have negligible impact on the utility of the models trained.

---

**Server-orchestrated training loop:**
  *parameters:* round participation fraction $q \in (0, 1]$, total number of users $N \in \mathbb{N}$, total number of rounds $T \in \mathbb{N}$, noise scale $z \in \mathbb{R}^+$, clip parameter $S \in \mathbb{R}^+$

  Initialize generator $\theta_G^0$, discriminator $\theta_D^0$, privacy accountant $\mathcal{M}$

  Set $\sigma = \frac{zS}{qN}$

  **for** each round $t$ from 0 to $T$ **do**
    $\mathcal{C}^t \leftarrow$ (sample of $qN$ distinct users)
    **for** each user $k \in \mathcal{C}^t$ **in parallel do**
      $\Delta_k^{t+1} \leftarrow \text{UserDiscUpdate}(k, \theta_D^t, \theta_G^t)$

    $\Delta^{t+1} = \frac{1}{qN} \sum_{k \in \mathcal{C}^t} \Delta_k^{t+1}$
    $\theta_D^{t+1} \leftarrow \theta_D^t + \Delta^{t+1} + \mathcal{N}(0, I\sigma^2)$

    $\mathcal{M}.\texttt{accum\_priv\_spending}(z)$

    $\theta_G^{t+1} \leftarrow \text{GenUpdate}(\theta_D^{t+1}, \theta_G^t)$

  print $\mathcal{M}.\texttt{get\_privacy\_spent}()$

**UserDiscUpdate$(k, \theta_D^0, \theta_G)$:**
  *parameters:* number of steps $n \in \mathbb{N}$, batch size $B \in \mathbb{N}$, disc. learning rate $\eta_D \in \mathbb{R}^+$, clip parameter $S \in \mathbb{R}^+$, gen. input size $n_U \in \mathbb{N}$, gen. function $G(U; \theta_G)$, disc. loss function $\ell_D(\theta_D; b_{\text{real}}, b_{\text{fake}})$

  $\theta_D \leftarrow \theta_D^0$
  $\mathcal{B} \leftarrow$ ($k$'s data split into $n$ size $B$ batches)
  **for** each batch $b_{\text{real}} \in \mathcal{B}$ **do**
    $U \leftarrow$ (sample $B$ random vectors of dim. $n_U$)
    $b_{\text{fake}} \leftarrow G(U; \theta_G)$  // *generated data*
    $\theta_D \leftarrow \theta_D - \eta_D \nabla \ell_D(\theta_D; b_{\text{real}}, b_{\text{fake}})$
  $\Delta = \theta_D - \theta_D^0$
  return update $\Delta_k = \Delta \cdot \min\left(1, \frac{S}{\|\Delta\|}\right)$  // *Clip*

**GenUpdate$(\theta_D, \theta_G^0)$:**
  *parameters:* number of steps $n \in \mathbb{N}$, batch size $B \in \mathbb{N}$, gen. learning rate $\eta_G \in \mathbb{R}^+$, gen. input size $n_U \in \mathbb{N}$, gen. loss function $\ell_G(\theta_G; b, \theta_D)$

  $\theta_G \leftarrow \theta_G^0$
  **for** each generator training step $i$ from 1 to $n$ **do**
    $U \leftarrow$ (sample $B$ random vectors of dim. $n_U$)
    $b_{\text{fake}} \leftarrow G(U; \theta_G)$  // *generated data*
    $\theta_G \leftarrow \theta_G - \eta_G \nabla \ell_G(\theta_G; b_{\text{fake}}, \theta_D)$
  return $\theta_G$

---

Algorithm 1: DP-FedAvg-GAN, based on DP-FedAvg (App. C) but accounts for training both GAN models.

Algorithm 1 ('DP-FedAvg-GAN') describes how to train a GAN under FL and DP. The discriminator update resembles closely the update in standard DP-FedAvg, and then each round concludes with a generator update at the server. The discriminator is explicitly trained under DP. During training the generator is only exposed to the discriminator, and never directly to real user data, so it has the same DP guarantees as the discriminator.

**Experiment**   Our second experiment shows how auxiliary DP federated GANs can be used to monitor an on-device handwriting classification network being used for inference. Once trained, the federated GANs synthesize privacy-preserving samples that identify the nature of a bug in on-device image preprocessing.

Consider the scenario of a banking app that uses the mobile phone's camera to scan checks for deposit. Internally, this app takes raw images of handwriting, does some pixel processing, and feeds the processed images to a pre-trained on-device Convolutional Neural Network (CNN) to infer labels for the handwritten characters. This CNN is the 'primary' model. The modeler can monitor its performance via metrics like user correction rate (i.e., how often do users manually correct letters/digits inferred by the primary model), to get coarse feedback on accuracy. To simulate users' processed data, we use the Federated EMNIST dataset (Caldas et al., 2018). Figure 2a shows example images; see Appendix C.1 for further details on the data.

Suppose a new software update introduces a bug that incorrectly flips pixel intensities during preprocessing, inverting the images presented to the primary model (Figure 2b).[4] This change to the primary model's input data causes it to incorrectly classify most handwriting. As the update rolls out to an increasing percentage of

---

[4]Such data issues are in fact common in practice. For example, the Federated EMNIST dataset we use encodes pixel intensities with the opposite sign of the MNIST dataset distributed by TensorFlow (TFF documentation, 2019).

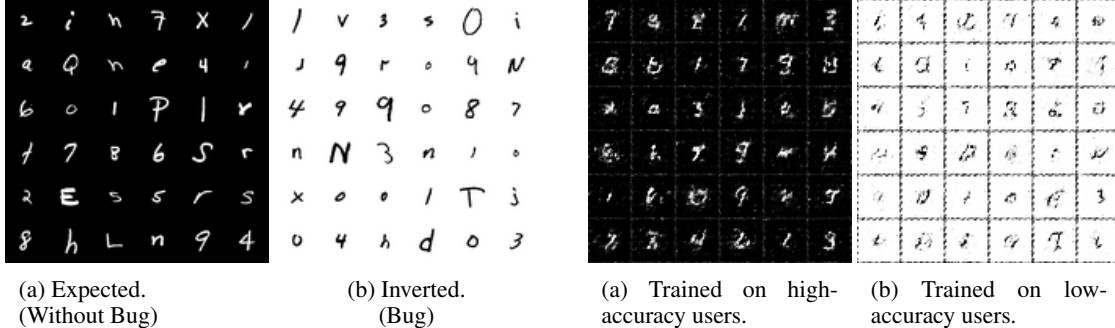

(a) Expected.
(Without Bug)

(b) Inverted.
(Bug)

(a) Trained on high-accuracy users.

(b) Trained on low-accuracy users.

Figure 2: Examples of primary model CNN input, from EMNIST with letters and digits (62 classes).

Figure 3: DP federated GAN generator output given an inversion bug on 50% of devices.

the app's user population, the user correction rate metric spikes, indicating to the app developers a problem exists, but nothing about the nature of the problem. Critically, the images in Figure 2 are never viewed by human eyes, but are fed to the primary model on device. If the data were public and the inference being performed on the app's server, some of the misclassified handwriting images could potentially be inspected, and the pixel inversion bug realized. But this cannot be done when the data is private and decentralized.

Instead, we train two DP federated GANs via Algorithm 1: one on a subset of the processed image data that tends to perform best when passed to the primary model, and one on a subset of processed image data that tends to perform worst. (Specifically, we do this by forming two distinct subpopulations of users who exhibit high and low classification accuracy, respectively, and training one GAN on each user subpopulation, as in **T4**; full details in Appendix C.3, and an alternative selection criteria explored in Appendix C.4.) By contrasting images synthesized by the two GANs, we hope to understand what is causing degraded classification accuracy for some app users. In our simulation of this scenario, we train for 1,000 communication rounds, with 10 users (drawn at random) participating in each round. See Appendix C.2 for training details. Figure 3 shows images generated by the two trained GANs, when the bug is present on 50% of users' phones.

Comparing the GANs' output, the presence of a pixel inversion bug is clear, enabling the modeler to locate and fix the issue.[5] A natural follow-on is to repeat training the two GANs after the fix is made; results shown in Figure 6 (Appendix C.3.1). Comparing these new images verifies the pixel inversion bug is fixed. It also potentially reveals new insights. Perhaps it identifies another, more subtle bug, or a particular characteristic of handwriting common to users with many misclassified examples (indicating a missing user class in the primary model's training data, i.e. an instance of **T6**). We propose it is useful to continually train federated generative models like the GANs in this experiment, as a tool to monitor and debug.

**Privacy at Scale** We follow an approach as in McMahan et al. (2018) to determine the level of privacy at realistic population sizes. We simulate with 10 users per round and a small total population size (3,400) in order to validate the utility of our models and also determine a noise scale multiplier $z$ that does not degrade the quality of the GAN's generated images. This simulation scenario will not afford good privacy protections (as indicated by the resulting very large $\epsilon$ bound). But it does indicate the factor necessary to scale $z$ up to 1.0; this is the factor we'd have to apply to per-round participation count to achieve a real-world scenario that affords good privacy protections (as measured by $\epsilon$ and $\delta$) while maintaining the quality of the GAN's generated images. As the experiment shows good utility is achieved with a $z$ of 0.01, the factor to apply is

---

[5]The relatively low quality of generated characters is irrelevant for this task; more training might result in higher quality, but these results are sufficient to diagnose the bug. Training can be stopped early for a better DP guarantee.

Table 3: Privacy parameters for different scenarios. $N$ is size of user subpopulation that meets selection criteria (not overall population size). Simulations are with overall population of 3,400, and realistic scenarios are with overall population of 2,000,000. All experiments use clip parameter $S$ of 0.1 and 1,000 rounds.

|  | $qN$ | $N$ | $z$ | $\epsilon$ | $\delta$ |
|---|---|---|---|---|---|
| simulation (Figure 3a) | 10 | 425 | 0.01 | $9.99 \times 10^6$ | $2.35 \times 10^{-3}$ |
| **realistic scenario** | $1,000$ | $250,000$ | 1.00 | 2.38 | $4.00 \times 10^{-8}$ |
| simulation (Figure 3b) | 10 | $2,125$ | 0.01 | $9.99 \times 10^6$ | $4.71 \times 10^{-4}$ |
| **realistic scenario** | $1,000$ | $1,250,000$ | 1.00 | 1.48 | $8.00 \times 10^{-9}$ |
| simulation (Figure 6) | 10 | 850 | 0.01 | $9.99 \times 10^6$ | $1.18 \times 10^{-3}$ |
| **realistic scenario** | $1,000$ | $500,000$ | 1.00 | 1.79 | $2.00 \times 10^{-8}$ |

100: a real-world use case would involve 1,000 users per round, out of a total population of e.g. 2 million mobile app users. This is reasonable for a real-world system, e.g. Bonawitz et al. (2019).

Table 3 gives DP hyperparameters and resulting $(\epsilon, \delta)$ values, both for the simulation and the corresponding scenario with scaled-up, realistic population. The calculation of $(\epsilon, \delta)$ is fully described in Appendix A.1. Note that three sets of numbers are presented, as $(\epsilon, \delta)$ depend on the number of users meeting the selection criteria for a given use case. The cases are: (i) high-accuracy users when the bug exists (12.5% of overall population), (ii) low-accuracy users when the bug exists (62.5% of overall population), and (iii) either group of users when the bug is gone (each 25% of overall population). In all cases, for the realistic population we achieve single digit $\epsilon$ (between 1.48 and 2.38), indicating a reasonable amount of user privacy.

## 7 CONCLUSION

In this work we described ML modeler workflows that rely on data inspection and are hence precluded when direct inspection is impossible (i.e., in a private, decentralized data paradigm). We proposed a tool to overcome this limitation: differentially private, federated generative models that synthesize examples representative of the private data. We demonstrated application of two example model classes (DP federated RNNs and GANs), showing how they enable a modeler to debug natural language and image problems. In the GAN case, we presented a novel algorithm for training under FL and DP. Section 2 discusses several other modeler workflows (e.g., debiasing, labeling) that rely on data inspection and are inhibited in a private, decentralized paradigm. Experiments applying DP federated generative models to these workflows is a promising direction for future work, especially since applications like data labeling will likely require higher-fidelity generative models than the ones considered here.

To convert the concept proposed in this paper into a practical data science tool, a variety of further research is necessary. For federated generative models to be useful and broadly applicable, they should require minimal tuning; excessive experimentation needed to achieve convergence both destroys the value proposition to the modeler and destroys the privacy budget under DP analysis. They must be capable of synthesizing relevant content in scenarios where the 'signal-to-noise' ratio is low, e.g., when a bug is only present in a small amount of data (in absolute or relative terms). How to achieve this in the case of GANs, which are powerful but can suffer from mode collapse, is of special interest. Advances in these dimensions will greatly benefit the private, decentralized data science community. As an initial step to stimulate research in this area, we provide an open-source implementation[6] of our DP federated GAN code (used to generate the results in Section 6). We share additional thoughts on open problems in Appendix D.

---

[6]https://github.com/tensorflow/federated/tree/master/tensorflow_federated/python/research/gans

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

# A    USER-LEVEL DIFFERENTIAL PRIVACY AND FEDERATED LEARNING

Differential privacy (DP) (Dwork et al., 2006; Dwork, 2011; Dwork & Roth, 2014) provides a well-tested formalization for the release of information derived from private data. Applied to ML, a differentially private training mechanism allows the public release of model parameters with a strong guarantee: adversaries are severely limited in what they can learn about the original training data based on analyzing the parameters, even when they have access to arbitrary side information. Formally, it says:

**Definition 1** *Differential Privacy: A randomized mechanism $\mathcal{M} \colon \mathcal{D} \to \mathcal{R}$ with a domain $\mathcal{D}$ (e.g., possible training datasets) and range $\mathcal{R}$ (e.g., all possible trained models) satisfies $(\epsilon, \delta)$-differential privacy if for any two **adjacent** datasets $d, d' \in \mathcal{D}$ and for any subset of outputs $S \subseteq \mathcal{R}$ it holds that $\Pr[\mathcal{M}(d) \in S] \leq e^\epsilon \Pr[\mathcal{M}(d') \in S] + \delta$.*

The definition above leaves open the definition of adjacent datasets which will depend on the application. Most prior work on differentially private ML (e.g. Chaudhuri et al. (2011); Bassily et al. (2014); Abadi et al. (2016); Wu et al. (2017); Papernot et al. (2017)) deals with *example*-level privacy: two datasets $d$ and $d'$ are defined to be adjacent if $d'$ can be formed by adding or removing a single training example from $d$.

However, recent work has presented an algorithm (DP-FedAvg, McMahan et al. (2018)) for differentially private ML with *user*-level privacy. Intuitively speaking, a model trained under DP-FedAvg should not change too much when one user's data is added, removed, or changed arbitrarily. Learning takes place via FL; since FL processes all of one user's data together, such guarantees are relatively easier to obtain than in the centralized setting.

We adopt an approach (Algorithm 2) for performing differentially private FL that follows closely from the DP-FedAvg algorithm. Similarly, our focus is on user-level differential privacy, where the privacy guarantee extends to all of a user's data (rather than e.g. a single training example).

Specifically, following DP-FedAvg, we:

- enforce clipping of per-user updates so the total update has bounded $L_2$ norm.

- use a bounded-sensitivity estimator for computing the weighted average update to the global model.

- add Gaussian noise to the weighted average update.

The sensitivity of a query is directly proportional to the amount of clipping $S$ and inversely proportional to the number of participants in a training round $qN$. Algorithm 2 shows that a noise scale parameter $z$ relates the amount of standard deviation $\sigma$ in the added Gaussian noise to this query sensitivity. As described in McMahan et al. (2018), acceptable DP bounds are typically achieved when $z \geq 1.0$. So a method of determining privacy hyperparameters that afford acceptable utility and privacy is to start with a small number of per-round participants $qN$, determine values of $S$ and $\sigma$ that provide good utility, and then increase the number of per-round participants so that $z \geq 1.0$.

A slight difference between our approach and that presented in DP-FedAvg is how we select devices to participate in a given federated round of computation. DP-FedAvg uses randomly-sized federated rounds, where users are selected independently with probability $q$. In this paper, we instead use fixed-size federated rounds, where the number of users selected to participate in the round, $qN$, is a constant (and $N$ is the total number of mobile devices participating in training). I.e., $q$ is the round participation fraction. This method of sampling has minor but non-trivial ramifications for the calculation of the overall privacy bound, which we discuss next.

---

**Server-orchestrated training loop:**
  *parameters*
    round participation fraction $q \in (0, 1]$
    total user population $N \in \mathbb{N}$
    noise scale $z \in \mathbb{R}^+$
    clip parameter $S \in \mathbb{R}^+$

  Initialize model $\theta^0$, privacy accountant $\mathcal{M}$

  Set $\sigma = \frac{zS}{qN}$

  **for** each round $t = 0, 1, 2, \ldots$ **do**
    $\mathcal{C}^t \leftarrow$ (sample without replacement $qN$ users from population)
    **for** each user $k \in \mathcal{C}^t$ **in parallel do**
      $\Delta_k^{t+1} \leftarrow \text{UserUpdate}(k, \theta^t)$

    $\Delta^{t+1} = \frac{1}{qN} \sum_{k \in \mathcal{C}^t} \Delta_k^{t+1}$
    $\theta^{t+1} \leftarrow \theta^t + \Delta^{t+1} + \mathcal{N}(0, I\sigma^2)$
    $\mathcal{M}.\texttt{accum\_priv\_spending}(z)$

  print $\mathcal{M}.\texttt{get\_privacy\_spent()}$

**UserUpdate$(k, \theta^0)$:**
  *parameters*
    number of local epochs $E \in \mathbb{N}$
    batch size $B \in \mathbb{N}$
    learning rate $\eta \in \mathbb{R}^+$
    clip parameter $S \in \mathbb{R}^+$
  *parameter functions*
    loss function $\ell(\theta; b)$

  $\theta \leftarrow \theta^0$
  **for** each local epoch $i$ from 1 to $E$ **do**
    $\mathcal{B} \leftarrow$ ($k$'s data split into size $B$ batches)
    **for** each batch $b \in \mathcal{B}$ **do**
      $\theta \leftarrow \theta - \eta \nabla \ell(\theta; b)$
  $\Delta = \theta - \theta^0$
  return update $\Delta_k = \Delta \cdot \min\left(1, \frac{S}{\|\Delta\|}\right)$   *// Clip*

---

Algorithm 2: DP-FedAvg with fixed-size federated rounds, used to train word- and char-LMs in Section 5.

## A.1 PRIVACY BOUNDS

To obtain precise DP guarantees, we use the analytical moments accountant of subsampled Rényi differential privacy (RDP) method developed in Wang et al. (2018). In particular, this work provides a tight upper bound on the RDP parameters for an algorithm that: (a) uniformly subsamples records from an underlying dataset, and then (b) applies a randomized mechanism to the subsample (Gaussian perturbation in our case). This is precisely what happens in one round of Algorithm 1 or 2. To track the overall privacy budget, we multiply the RDP orders of the uniformly subsampled Gaussian mechanism by the number of rounds we execute Algorithm 1 or 2 and convert the resulting RDP orders to an $(\epsilon, \delta)$ pair using Proposition 3 of Mironov (2017).

## A.2 ADDITIONAL CONSIDERATIONS

Note that $(\epsilon, \delta)$ DP upper bounds are often loose, and so can be complemented by direct measurement of memorization, as in Carlini et al. (2018); Song & Shmatikov (2019). Further, while often unrealistic as a threat model, membership inference attacks on GANs can also be used as a tool to empirically quantify memorization (Hayes et al., 2019; Hilprecht et al., 2019). Finally, we note that since the modeler has access to not only the auxiliary generative models, but also potentially other models trained on the same data, if an $(\epsilon, \delta)$ guarantee is desired, the total privacy loss should be quantified across these models, e.g. using the strong composition theorem for DP (Mironov, 2017; Kairouz et al., 2017).

## B    DP Federated RNNs: Experimental Details

### B.1    Stack Overflow data

The Stack Overflow dataset contains questions and answers from the Stack Overflow forum grouped by user ids. The data consists of the text body of all questions and answers. The bodies were parsed into sentences; any user with fewer than 100 sentences was expunged from the data. Minimal preprocessing was performed to fix capitalization, remove URLs etc. The corpus is divided into train, held-out, and test parts. Table 4 summarizes the statistics on number of users and number of sentences for each partition. The dataset is available via the Tensorflow Federated open-source software framework (Ingerman & Ostrowski, 2019).

Table 4: Number of users and sentences in the Stack Overflow dataset.

|  | Train | Held-out | Test |
|---|---|---|---|
| # Users | 342K | 38.8K | 204K |
| # Users with both questions and answers | 297K | 34.3K | 99.3K |
| # Sentences | 136M | 16.5M | 16.6M |
| # Sentences that are questions | 57.8M | 7.17M | 7.52M |
| # Sentences that are answers | 78.0M | 9.33M | 9.07M |

### B.2    RNN Model Architectures and Training

**DP Word-LM**    The DP word-LM model is based on a coupled input forget gate CIFG-LSTM (Greff et al., 2015) with projection layer (Sak et al., 2014). It uses a 10K vocabulary, which is composed of the most frequent 10K words from the training corpus. The model architecture is based on an RNN with hidden size of 670 and an input embedding dimension of 96. The model is trained for 2,000 rounds. A server learning rate of 1.0, an on-device learning rate of 0.5, and Nesterov momentum of 0.99 are used.

**DP Char-LM**    The model architecture used to train the DP char-LM is based on "FL$_\mathrm{L}^\mathrm{M}$" in Chen et al. (2019), which is also a CIFG-LSTM with projection using 3 layers, 256 RNN hidden units and 128 projected dimensions. The input embedding dimension is also 128. The character vocabulary has a size of 258, based on UTF-8 encoding with 256 categories from a single byte and additional start-of-word and end-of-word tokens. All other training and optimization parameters are the same as the word-LM model. Similar to Chen et al. (2019), once the char-LM is trained, Monte Carlo sampling is performed to generate OOV words.

**Privacy Hyperparameters**    Table 5 gives the privacy hyperparameters used with DP-FedAvg (Algorithm 2) to yield the trained word-LM and char-LM models that generated the text results presented here.

Table 5: Privacy hyperparameters for DP RNN experiments.

|  | $L_2$ clip parameter $S$ | participation count $qN$ | total users $N$ | noise scale $z$ | number of rounds $T$ |
|---|---|---|---|---|---|
| DP word-LM | 0.2 | 5,000 | 342,777 | 1.0 | 2,000 |
| DP char-LM | 0.1 | 5,000 | 342,777 | 1.0 | 2,000 |

### B.3    EXPANDED RESULTS

To demonstrate the effectiveness of using the OOV model to monitor the introduced system bug, the experiments are done in four comparison settings: (1) without bug; (2) 1% of sentences have first two words concatenated; (3) 10% sentences; (4) 100% sentences. In each setting the model trains for 2,000 communication rounds, to meet the privacy guarantees stated in Section 5. We observed in all four settings that models were converged after 2,000 rounds.

**DP Word-LM**    Figure 4 shows percentage of OOV words by their position in the sentence. It clearly shows that as the percentage of sentences affected increases, the word-LM reflects this in its generated output.

Table 6 shows the overall OOV rate observed in four experiment settings. It can be seen that overall OOV rate goes up as the percentage of sentences affected by the concatenation bug increases. Thus overall OOV rate can be used as an early abnormality indicator of the corpus.

Table 7 shows samples of phrases generated from the word-LM in different experiment settings. We can see that the phrases have more OOVs (marked as UNK) as the percentage of sentences affected by the concatenation bug increases.

**DP Char-LM**    Table 8 shows the 20 most frequently occurring OOV words generated by the char-LM, for the four different experiment cases. These generated results provide actual information on the tokenized words being fed to the model from the decentralized dataset. Similar to the word-LM, as the ratio of the bug increases, the chance of generating content reflecting the bug (in this case, concatenated words) becomes higher. In the 100% setting, all the top 20 words are concatenated words; the word-level joint probabilities of the concatenated words are significantly higher than their non-concatenated counterparts.

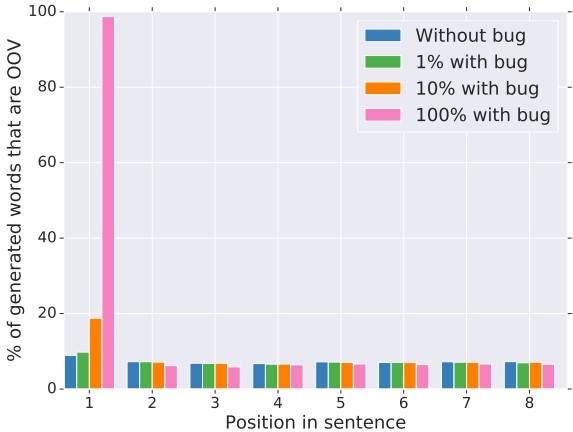

Figure 4: Percentage of samples generated from the word-LM that are OOV by position in the sentence (computed over 100,000 samples).

Table 6: Overall OOV rate observed (during training) in different word-LM experiment settings.

| without bug | 1% | 10% | 100% |
|---|---|---|---|
| 6.52% | 6.63% | 7.60% | 17.89% |

Table 7: Sample phrases generated from word-LMs when 0%, 1%, 10%, and 100% of sentences affected.

| *without bug* | *1%* |
| --- | --- |
| for UNK metrics to work fine , ago | this tries to load the textbox by get |
| i think your question is the correct format | if any user who was originally have an |
| question 1 UNK ? java 1 , UNK | which one is different at the same time |
| is the selector of the type of module | UNK currently building my ios app to integrate |
| basically , i do a bunch of articles | the only thing i can think of is |
| UNK data UNK UNK UNK @ 43 UNK | / home / UNK / jmeter / ; |
| this is a blog post for huge UNK | the solution with our application is the current |
| one possibility is to send different limits to | there is another solution in their UNK feed |
| this is missing some functions UNK on index | ( your client sends all the source properties |
| after that UNK start the above , then | thank you .  .  .  in your worst |

| *10%* | *100%* |
| --- | --- |
| looks like it are pretty weather when i | UNK lg for this is computing 3 as |
| suppose one key is executed when you freeze | UNK it is that sql will stop over |
| delete disadvantage of the files and prediction . | UNK of UNK you need to call UNK |
| UNK a folder created on one system ( | UNK ms might have this off ; & |
| please note that the button would mount a | UNK in the choice of sort of dragging |
| would it be displaying the same field i | UNK , p1 .  UNK ( UNK , |
| UNK just all the UNK files , such | UNK UNK editing the push ( ) controllers |
| UNK , it UNK matter why ( note | UNK info must have depend on the way |
| simply at UNK . UNK but put it | UNK i do it static ?  UNK stop |
| i started many debug from the jboss | UNK trying to give more html privileges ? |

Table 8: Top 20 char-LM-generated OOV words by joint character probability, for 0% (without bug), 1%, 10%, and 100% of examples affected by bug. Value alongside is joint probability (computed via Equation 1).

| *without bug (prob $10^{-3}$)* | | *1% (prob $10^{-3}$)* | | *10% (prob $10^{-3}$)* | | *100% (prob $10^{-3}$)* | |
|---|---|---|---|---|---|---|---|
| regex | 5.50 | regex | 5.15 | i have | 8.08 | i have | 27.47 |
| jsfiddle | 3.90 | jsfiddle | 3.73 | i am | 5.60 | i am | 16.50 |
| xcode | 3.12 | xcode | 2.86 | regex | 4.45 | this is | 9.92 |
| divs | 2.75 | listview | 2.64 | you can | 3.71 | you can | 9.04 |
| stackoverflow | 2.75 | textbox | 2.56 | if you | 2.81 | here is | 8.30 |
| listview | 2.74 | divs | 2.45 | this is | 2.77 | if you | 8.29 |
| textbox | 2.73 | stackoverflow | 2.44 | here is | 2.73 | i want | 8.17 |
| foreach | 2.34 | iis | 2.27 | jsfiddle | 2.70 | is there | 7.51 |
| async | 2.27 | foreach | 2.24 | i want | 2.39 | how can | 5.78 |
| iis | 2.21 | linq | 2.20 | textbox | 2.28 | when i | 5.33 |
| onclick | 2.07 | async | 2.17 | xcode | 2.23 | however , | 5.25 |
| linq | 2.02 | onclick | 2.05 | listview | 2.10 | i would | 4.72 |
| params | 2.00 | htaccess | 1.90 | stackoverflow | 2.10 | if i | 4.22 |
| htaccess | 1.90 | params | 1.89 | is there | 2.07 | for example | 4.20 |
| arraylist | 1.90 | mongodb | 1.86 | divs | 1.93 | i tried | 4.14 |
| mongodb | 1.83 | npm | 1.77 | async | 1.79 | edit : | 4.07 |
| xaml | 1.79 | xaml | 1.74 | linq | 1.72 | i think | 3.86 |
| npm | 1.66 | arraylist | 1.67 | foreach | 1.71 | the problem | 3.77 |
| dataframe | 1.61 | enum | 1.58 | iis | 1.70 | i don't | 3.61 |
| nginx | 1.55 | nginx | 1.55 | however , | 1.69 | i need | 3.49 |

## C  DP FEDERATED GAN: EXPERIMENTAL DETAILS

### C.1  FEDERATED EMNIST DATA

We use the Federated EMNIST dataset (Caldas et al., 2018) to represent the app's users' processed data in our GAN experiments in Section 6. This dataset contains 28x28 gray-scale images (Figure 2a) of handwritten letters and numbers, grouped by writer. This has the realistic property that all data grouped to a given user exhibits the same personally-identifying handwriting properties; our solution should generate examples while obfuscating these properties. The dataset contains 3,400 users. It is available via the Tensorflow Federated open-source software framework (Ingerman & Ostrowski, 2019).

### C.2  FEDERATED GAN ARCHITECTURE AND TRAINING

**Models and Losses**  Our generator and discriminator network architectures are borrowed from a popular GAN code tutorial[7] for MNIST. For the generator and discriminator loss functions, $l_G()$ and $l_D()$, we used the Improved Wasserstein loss formulation presented in Gulrajani et al. (2017). We used a gradient penalty coefficient of 10.0.

**Hyperparameters**  Table 9 gives the various hyperparameters used with the DP-FedAvg-GAN algorithm to yield generators that produced the images displayed in Figures 3, 6, 7, and 8.

---

[7]https://github.com/tensorflow/gan

Table 9: DP federated GAN experimental hyperparameters.

| generator training steps $n$ | generator batch size $B$ | generator learning rate $\eta_G$ | generator input size $n_U$ |
|---|---|---|---|
| 6 | 32 | 0.005 | 128 |
| discriminator training steps[8] $n$ | discriminator batch size[8] $B$ | discriminator learning rate $\eta_D$ | $L_2$ clip parameter $S$ |
| $\leq 6$ | $\leq 32$ | 0.0005 | 0.1 |
| participation count $qN$ | total users $N$ | noise scale $z$ | number of rounds $T$ |
| 10 | *see Table 11* | 0.01 | $1,000$ |

**Network Update Cadence**    For simplicity, Algorithm 1 shows a 1:1 ratio of discriminator:generator training updates. This is not required. The discriminator could go through multiple federated updates before an update to the generator is calculated, just as is the case with non-federated GANs (Goodfellow et al., 2014). In these experiments we used a 1:1 ratio of discriminator:generator training updates, as this update cadence empirically was determined to work well.

### C.3 SELECTION CRITERIA: FILTERING BY USER

The modeler ideally wishes to train one GAN on the exact subset of data affected by the bug and the other GAN on the exact subset of data unaffected by the bug, in order to best discern the difference and identify the bug. Of course, to achieve this perfectly they would need to know what the actual bug is a priori. As they don't, the modeler must settle for focusing the two GANs to train on the subsets of the data with the highest and lowest likelihood (respectively) of being related to the bug.

A question is how to define these two subsets of the overall decentralized dataset. In the experiments in Section 6, we filter these two subsets at the level of the *user* (i.e., **T4** in Table 1). We define criteria for a user's app experience to be considered as 'poor', and if a user meets this criteria then their device belongs to the 'poorly performing' subpopulation. When their device is selected to participate in a round, their examples will be used to calculate updates to the GAN discriminator for this subpopulation. (The equivalent is true for identifying members of the 'strongly performing' subpopulation.)

The criteria we use is a user's classification accuracy, i.e., the percentage of examples in their local dataset that were correctly classified by the primary model (the CNN). Figure 5 shows a histogram of classification accuracy for the app's users, and Table 10 gives the 25th and 75th percentiles when no bug exists. We set accuracy thresholds of 'poor' (and 'strong') performance based on the 25th (and 75th) percentile, i.e., when no bug exists we aim to capture the 25% of users who experience the worst (and best) classification accuracy.

When a bug exists and performance of the primary model drops, the 'poorly performing' subpopulation swells and the 'strongly performing' subpopulation shrinks (note the shift in histogram in Figure 5 with and without the bug). That is, subpopulation size varies depending on the degree of impact of the bug. Table 11 gives the subpopulation sizes that resulted for the experiments we ran.

---

[8]Each user in the Federated EMNIST set has a different number of examples. Some users have less than $6 \times 32$ examples total. When such a user is selected for a round by the coordinating server, the discriminator update will use all its examples to train, but the batch size of the final batch could be less than 32 and number of steps taken could be less than 6.

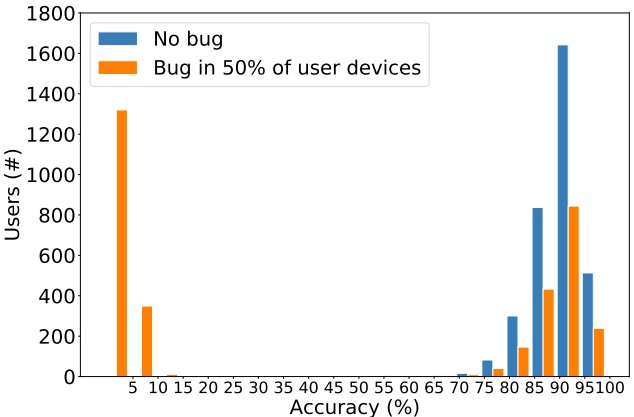

Figure 5: User accuracy histogram.

Table 10: User accuracy percentiles (without bug).

|                          | Accuracy |
|--------------------------|----------|
| 75th Percentile of Users | 93.9%    |
| 25th Percentile of Users | 88.2%    |

Table 11: Subpopulation sizes for simulations in Section 6. Total population size is 3,400 user devices.

|                      | Without Bug | Bug in 50% of user devices |
|----------------------|-------------|----------------------------|
| High-Accuracy Users  | 850         | 425                        |
| Low-Accuracy Users   | 850         | 2,125                      |

Of course, this is not a perfect manner for identifying which data contains the bug (although as the bug is unknown, no manner could be perfect a priori). The poorly performing subpopulation will always contain the 25% of the overall population that doesn't classify well for reasons unrelated to any bug (e.g., users with irregular handwriting, etc.), and their data will be training the discriminator along with the data actually affected by the bug. The assumption is that either the generative model has the capacity to model both modes of data, or the unaffected data will be dwarfed by the affected data and the generative model will focus on the larger mode of data. This assumption held for this experiment, but further research on the 'sensitivity' of selection criteria is necessary (see Section 7 and Appendix D).

An alternative selection criteria, where data is split up by classified and misclassified examples (as opposed to by high accuracy and low accuracy users), is presented in Appendix C.4.

### C.3.1 ADDITIONAL GENERATED RESULTS

Figure 6 shows the output of two DP federated GANs trained on the best and worst-classifying devices, once the bug described in Section 6 is fixed. These synthetic images are of high-enough quality to verify to the modeler that the pixel inversion bug they observed in Figure 3 is gone.

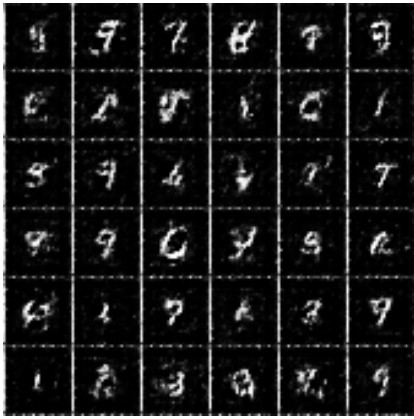
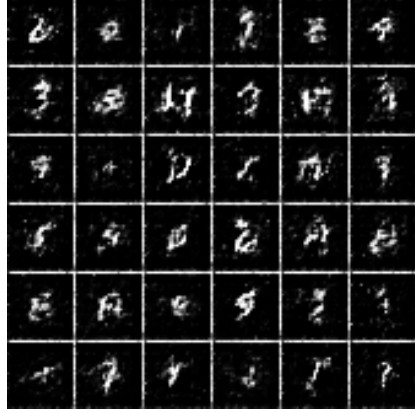

(a) Trained on high-accuracy users.                    (b) Trained on low-accuracy users.

Figure 6: DP federated GAN generator output after the inversion bug is fixed.

### C.4 ALTERNATIVE SELECTION CRITERIA: FILTERING BY EXAMPLE

As Section 2 describes, there are a number of selection criteria that an ML modeler might employ in the course of their modeling tasks. The experiment described in Section 6 and Appendix C.3 involved data selection that filtered 'by user', based on a user's classification accuracy (i.e., **T4** in Table 1). Here we demonstrate an alternative selection criteria: filtering 'by example', based on whether a data example was correctly classified by the primary model (i.e., **T2** in Table 1). We show that we can equivalently use this alternative data selection and train DP federated GANs to debug the primary model and uncover the source of the drop in accuracy observed in Figure 5.

Informally, one can consider the difference between filter 'by user' and filter 'by example' as "train on all examples, for some users" vs. "train on some examples, for all users". In the former, the amount of users contributing to training varied quite a bit based on whether the bug was present or absent and whether we were training the high-accuracy or low-accuracy subpopulation (Table 11). With the latter, the amount of

Table 12: Subpopulation sizes for filter 'by example' simulations, when excluding a user's participation in subpopulation if containing $< 5$ examples of classification result. Total population size is 3,400 users.

|  | Without Bug | Bug in 50% of user devices |
| --- | --- | --- |
| Correctly Classified Examples | 3,400 | 2,905 |
| Misclassified Examples | 3,282 | 3,340 |

users in a training subpopulation varies to a much smaller extent (Table 12). Some variation in subpopulation sizes is still present, due to our requiring a user device to contain a minimum number of examples of a classification result (either 'correctly classified' or 'misclassified') in order for it to participate in training for that particular subpopulation. This minimum number of examples threshold is applied to mitigate effect of outliers. In the experiment we used a threshold of 5.

**Experiment**   As in Section 6, we train two distinct DP federated GANs via Algorithm 1, in order to infer the nature of the bug from the contrast in synthesized content between the two. One GAN trains on the examples on each device that were correctly classified by the primary model, and the other trains on examples that were misclassified. We again trained each GAN for 1,000 communication rounds involving 10 users per round.

Figures 7a and 7b show examples of images generated by the two GANs when trained with the bug present on 50% of users' mobile phones. As with the 'by user' experiment, the 'by example' results here show that DP federated GANs can be trained to high enough fidelity to clearly distinguish that some private data has black and white reversed. Figures 8a and 8b show examples of images generated by the two GANs when trained with the bug present on no phones (e.g., after the bug is fixed).

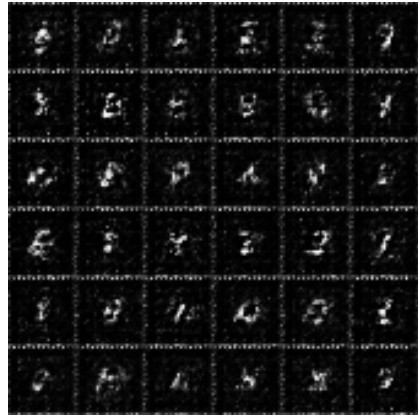
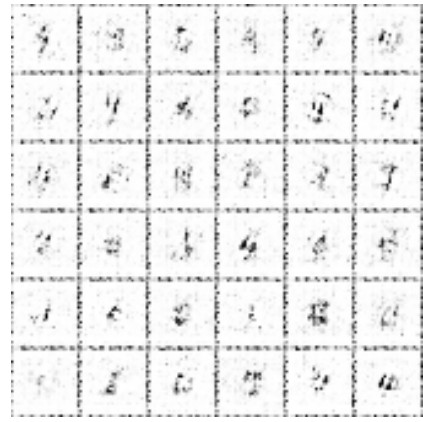

(a) Trained on correctly classified examples (for all users).

(b) Trained on misclassified examples (for all users).

Figure 7: Filter 'by example' simulation: generator output with the inversion bug present on 50% of devices.

**Privacy at Scale**   Privacy bounds for these 'by example' experiments are determined in analogous fashion to the 'by user' experiments in Section 6. In simulation we ran with 10 users per round and total population size of 3,400, to validate good utility in the presence of clipping and noising. While the $\epsilon$ and $\delta$ bounds

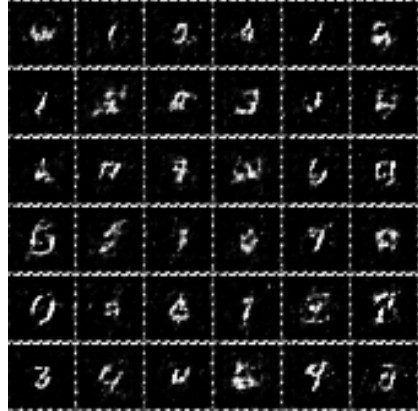

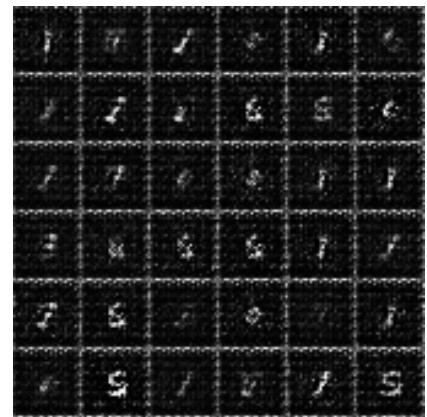

(a) Trained on correctly classified examples (for all users).

(b) Trained on misclassified examples (for all users).

Figure 8: Filter 'by example' simulation: generator output after the inversion bug is fixed.

Table 13: Privacy parameters for different filter 'by example' (**T2**) scenarios. $N$ is size of user subpopulation that meets selection criteria (not overall population size). Simulations are with overall population of 3,400, and realistic scenarios are with overall population of 2,000,000. All experiments use clip parameter $S$ of 0.1 and 1,000 rounds.

|  | $qN$ | $N$ | $z$ | $\epsilon$ | $\delta$ |
|---|---|---|---|---|---|
| simulation (Figure 7a) | 10 | 2,905 | 0.01 | $9.99 \times 10^6$ | $3.44 \times 10^{-4}$ |
| **realistic scenario** | 1,000 | 1,708,824 | 1.00 | 1.47 | $5.85 \times 10^{-9}$ |
| simulation (Figure 7b) | 10 | 3,340 | 0.01 | $9.99 \times 10^6$ | $2.99 \times 10^{-4}$ |
| **realistic scenario** | 1,000 | 1,964,706 | 1.00 | 1.40 | $5.09 \times 10^{-9}$ |
| simulation (Figure 8a) | 10 | 3,400 | 0.01 | $9.99 \times 10^6$ | $2.94 \times 10^{-4}$ |
| **realistic scenario** | 1,000 | 2,000,000 | 1.00 | 1.39 | $5.00 \times 10^{-9}$ |
| simulation (Figure 8b) | 10 | 3,282 | 0.01 | $9.99 \times 10^6$ | $3.05 \times 10^{-4}$ |
| **realistic scenario** | 1,000 | 1,930,588 | 1.00 | 1.40 | $5.18 \times 10^{-9}$ |

do not guarantee meaningful privacy protection at this small scale of participants, the amount of clipping and noising are such that if we to scale up to a larger realistic scenario, we have meaningful DP guarantees. Table 13 summarizes the privacy bounds, and shows we achieve single digit $\epsilon$ bounds for realistic scenarios involving 1,000 users per round and total population size of 2,000,000 user devices. (For each realistic scenario, we estimated the number of devices $N$ in the user subpopulation by taking the same proportion of users out of the overall population as occurred in the corresponding simulation scenario.) Reiterating from Section 6, this scale of orchestration is feasible for real-world production FL systems, e.g. Bonawitz et al. (2019).

## D   OPEN PROBLEMS

The intent of this paper is to highlight the gaps in a modeler's data inspection toolbox when undertaking private, decentralized learning, and to present what appears to be a promising solution direction involving generative models. To say that open work remains is an understatement, and there are many interesting topics to consider in this new space.

**Alternative Notions of Privacy**   In this paper we implemented user-level privacy via the mechanism of $(\epsilon, \delta)$-differential privacy (DP), and considered acceptable privacy to be achieved via single-digit bounds on $\epsilon$. Other useful notions of privacy exist, e.g. Carlini et al. (2018) defines a concept called *exposure*, a more empirical measure of data disclosure that takes into account the network training process. Research like Carlini et al. (2018) and Jayaraman & Evans (2019) shows that for many problems a large gap exists between empirical measures and DP's measure of privacy risk. While DP is tighter, empirical concepts like exposure may prove to be more practically useful for a given application; even if the $\epsilon$ bound is larger than single-digits, *exposure* can indicate that data memorization is not taking place and privacy protected. Exploring how alternative concepts of privacy affect the utility of federated generative models could improve their usefulness. (This topic connects with the FedGP work in Triastcyn & Faltings (2019) exploring differential *average-case* privacy for federated GAN training.)

**Sensitivity**   The GAN experiment in Section 6 demonstrated debugging when 50% of user devices have been affected. The RNN experiment in Section 5 demonstrated debugging when 10% of examples have been affected. In Appendix B we present RNN results with differing percentages of affected examples, but in general a question remains: how sensitive to the presence of a bug are such federated generative models? Is there a minimum percentage of users or examples that's required in a composite data distribution such that the model learns to generate examples from that part of the distribution? This is of particular interest in the case of GANs, which are known to suffer from problems of mode collapse (Salimans et al., 2016).

**Federated GAN Training**   We chose GANs for this paper as they had not (to our knowledge) previously been trained with FL and DP, and developing an algorithm to do so was an open question. Apart from the DP-FedAvg-GAN algorithm we present, the GAN used is relatively simplistic (see details in Appendix C.2). Exploring the space of GAN techniques to study which losses and model architectures are most amenable to privatization (e.g., scaling and noising for DP) and federation would be immensely useful.

**Other Classes of Generative Models**   Other types of generative models exist, with interesting questions on how to federate them. For example, Variational Autoencoders (VAEs) involve an encoder mapping high-dimensional data examples to a lower-dimensional latent space, and a decoder mapping from latent space back to high-dimensional space. With decentralized user-private data, is a useful paradigm to have each user have a personalized, private encoder (specific to the particulars of their local data and how it maps to the latent space), with a common decoder serving as synthesis engine mapping latent space to examples?

**Federating Contrastive or Triplet Losses**   A class of ML losses involve distinguishing examples (or users) from each other. An example is the triplet loss used for FaceNet in Schroff et al. (2015). The loss is the sum of a reward based on similarity to a positive example and a penalty based on similarity to a negative example. For such loss calculations, a compute node requires access to both positive and negative examples.

FL protects not only the privacy of users from the coordinating server, but also the privacy of users from each other; each user's data resides in a distinct silo. Computing a triplet loss on a user's device requires access to negative examples of data (i.e., from other users), which is not directly possible. DP federated generative models offer a potential solution. If one can be trained to embody a user population, it can then be shipped to users' devices to serve as an engine providing synthetic negative examples for the triplet loss calculation.

