# OpenReview forum: "Generative Models for Effective ML on Private, Decentralized Datasets"
_ICLR.cc/2020/Conference — Accept (Poster)_

### Official Review · AnonReviewer3 · 2019-10-24
**Official Blind Review #3**

**Rating:** 3

**Review:**

This paper proposes a differentially private federated learning method to learn GAN with application to data bugging situations where privacy protection is needed. The proposed method tries to leave the data at the user-end to train the discriminators, and learn the generator at the centralised server. To support the debugging data related issues as claimed, two specific examples related to text and image modeling were presented. It is the generator which is DP-protected (as the discriminators are DP-protected) makes it possible where the generated data can hint the potential bugs.

The scenario being considered is interesting and two real examples have used to illustrate the idea. However, this paper falls short in the following ways:
- It adopts what being proposed in McMahan et al. (2018) with some modifications to achieve the goal. The novelty is more related to the proposed application which allows debugging data issues to be possible when the data is private and decentralised.
- The two debugging illustrations are very specific in term of the errors introduced and the ways to achieve the debugging goal. It is not sure how they can be further generalized to other types of bugs.
- The paper is well written. However, the readers should have reasonable background on DP, GAN, federated learning, and generative models, or it will be hard to read through. Having said that, the authors do provide quite comprehensive literature review on related topics. But, then not much space is left for providing the necessary background and details for  the proposed federated learning for GAN with DP (other than referring to Algorithm 1). The experiment section is good.

Specific questions:
- Other than the tokenisation bug and the image insertion bug, can more possible examples be described?
- Can the examples be generalised into some methodologies? And, what are the limits? Will there be data inspection needs which cannot be achieved by this approach? What are they?

**Experience Assessment:**

I have published one or two papers in this area.

**Review Assessment: Checking Correctness Of Derivations And Theory:**

I assessed the sensibility of the derivations and theory.

**Review Assessment: Checking Correctness Of Experiments:**

I assessed the sensibility of the experiments.

**Review Assessment: Thoroughness In Paper Reading:**

I read the paper at least twice and used my best judgement in assessing the paper.

---

> ### Author Response · Authors · 2019-11-09
> **Response to Reviewer #3**
>
> We thank the reviewer for their comments.  We answer their specific questions in turn.
>
> Question #1 (Re: generality of approach to bugs, choice of bugs for paper)
>
> We respectfully disagree with the reviewer’s conclusions on the generality of this approach (e.g., “The two debugging illustrations are very specific in term of the errors introduced and the ways to achieve the debugging goal. It is not sure how they can be further generalized to other types of bugs.”), though we acknowledge this generality was not sufficiently described in our initial submission. We have significantly revised Section 2, including adding a new Section 2.1, which we hope resolves these concerns.  In particular, apart from selecting a type of generative network that best applies to the problem domain (e.g., choosing RNNs to debug a language modeling task, choosing GANs to debug an image modeling task), no further assumptions need be made by the user about the nature of the data (or any bugs or biases therein).
>
> Indeed, it is precisely because the signal we are trying to detect is unknown that we recommend a generative model.  Were the modeler to possess additional evidence that strongly indicated a particular type of bug, a simpler data analysis may be enough to detect it.  (E.g., if the modeler of the image pipeline had reason to strongly suspect a priori that some user’s images were black/white inverted, they could instead compute per-user device average pixel values and use federated computation to aggregate into a histogram.  The histogram would reveal that many devices had predominantly very white images.)  But because we assume no such a priori knowledge, we desire an approach that is as general as possible.
>
> Note that in Section 2 we now reference a recently published survey paper providing a taxonomy of faults in deep learning systems (https://arxiv.org/abs/1910.11015).  We feel this paper confirms our choice of bugs as being representative examples, as they are listed prominently (e.g., ‘text segmentation’, ‘pixel encoding’) under one of the largest subcategories of faults, ‘Preprocessing of Training Data’  (https://arxiv.org/abs/1910.11015, Figure 1).
>
>
> Question #2 (Re: can approach be generalized into some methodologies, what are the limits)
>
> As mentioned, we have attempted to make the general methodology prominent in Section 2.
>
> Assessing the limits of this approach is one of the most interesting questions to explore in future work.  To make an analogy: In this paper we present the use of a new ‘sensor’ and show its promise; we hope to see the community take up research on this sensor so we can all work together to characterize its ‘signal-to-noise’ ratio.
>
> Some different limits that can be thought of are sensitivity (i.e., how much presence in the underlying data distribution is required before representative examples of a characteristic start to be synthesized) and fidelity (i.e., how realistic do synthesized examples need to be to detect a characteristic).  We’ve attempted to discuss each in the paper; we welcome the reviewer’s feedback if the current discussion in the paper could be improved.
>
> Sensitivity : We empirically characterize the sensitivity limits of the approach in the paper, e.g., in Figure 4 and Table 4.  There we show, for RNN language models trained with varying degrees of presence of the concatenation bug, the varying levels of UNKs noticed in the generated content.  E.g., Figure 4 shows that when the bug is only present in 1% of sentences, the distribution of generated content is close to unchanged vs. the no-bug case; however, when the bug is present in 10% of sentences, a clear change in distribution of UNKs is noticeable.  Does the reviewer feel this a useful empirical analysis of sensitivity limits?
>
> Fidelity : Section 2.1 now contains a discussion of the types of problems where lower-fidelity synthesis is ok and the problems where high-fidelity will likely be necessary. Please let us know if this addresses the reviewer’s concerns.  (Thanks to this reviewer and other reviewers’ feedback, as we realized this matter was less clearly discussed in the initial draft.)
>
> Again, we hope this paper encourages new work in the generative modeling community, in particular to both assess current limits and hopefully push them further.  (Towards this end, we call attention to open questions about fidelity and sensitivity in the Conclusion and Open Problems sections, respectively.)  But we feel this initial paper demonstrates that there are realistic data inspection problems that exist today that can already be addressed with an approach like we describe, i.e., are useful within current limits of sensitivity and fidelity.

---

### Official Review · AnonReviewer2 · 2019-10-24
**Official Blind Review #2**

**Rating:** 6

**Review:**

This work presents a method for using generative models to gain insight into sensitive user data, while maintaining guarantees about the privacy of that data via differential privacy (DP) techniques. This scheme takes place in the federated learning (FL) setting, where the data in question remains on a local device and only aggregate updates are sent to a centralized server. The intended application here is to use the trained generative models as a substitute for direct inspection of user data, thus providing more tools for debugging and troubleshooting deployed models in a privacy conscious manner.

Pros:
Given the growing computational power of mobile devices and the importance of privacy for large-scale deployment of machine learning, this work is a timely contribution that could augment the ML pipeline for at-scale applications dealing with sensitive data. The authors do a good job of fleshing out the intended use cases of their training scheme, and present a pair of experiments that are well-chosen for illustrating the utility of generative models when dealing with private data.

Cons:
Although likely of practical use, the work seems to be lacking in novelty in several respects. First, the techniques developed here represent a fairly straightforward merger of DP and FL tools without much in the way of qualitatively new offerings. While the authors do develop a new GAN training scheme that works in the FL setting, this adaptation is also pretty straightforward, and mostly follows the approach laid out in [1] for training recurrent neural nets.

Secondly, this paper comes in the midst of many other works aiming to integrate different combinations of generative models, privacy, and distributed training (as pointed out in the related work section). While the particular combination of techniques here differ from those in previous work, the authors don't attempt to justify why their training scheme should be preferred over these prior methods. And although their experiments are useful for understanding the general utility of generative models trained in a private and decentralized setting, they unfortunately don't permit any direct comparison with the experiments used in these previous papers.

Verdict:
For the reasons given above, I cannot recommend acceptance of this work.

[1] H Brendan McMahan, Daniel Ramage, Kunal Talwar, and Li Zhang, Learning differentially private recurrent language models, ICLR 2018

*** Follow-up after authors' rebuttal ***

I'd like to thank the authors for their rebuttal, and for the significant addition to the paper in the form of an expanded Section 2. This material has helped me gain a bit better perspective on the use cases for their work, and convinced me of the potential for their methodology within real-world development of deep learning tools and services. In addition, this additional context helps to motivate the two experiments described here as fair representatives of actual debugging problems, and not simply issues that were hand-chosen to prove the authors' point.

I still hold that the paper offers very little in the way of new conceptual or technical contributions, but in light of the potential utility of this privacy-conscious generative pipeline for the broader deep learning community, I have changed my score from a weak reject to a weak accept.

**Experience Assessment:**

I do not know much about this area.

**Review Assessment: Checking Correctness Of Derivations And Theory:**

N/A

**Review Assessment: Checking Correctness Of Experiments:**

I assessed the sensibility of the experiments.

**Review Assessment: Thoroughness In Paper Reading:**

I read the paper at least twice and used my best judgement in assessing the paper.

---

> ### Author Response · Authors · 2019-11-09
> **Response to Reviewer #2**
>
> We appreciate the reviewer’s comments.
>
> First, we wish to clarify our view of the contributions of the paper. The principle contribution is not the introduction of new algorithms, but of a methodology for combining existing techniques together with a careful selection procedure in order to solve a large set of ML modeling challenges when working with decentralized data. While this observation may seem straightforward in hindsight, we do not believe it has been presented in any previous works. We have revised Section 2 and added a new Section 2.1 which hopefully makes this contribution more clear. While indeed we did need to make some algorithmic contributions (training user-level DP GANs on decentralized data for the first time), this is a secondary contribution.
>
> We think something that was missed in the initial review of our paper was the uniqueness of combining federated learning, generative models, and user-level differential privacy.  We respectfully disagree with the reviewer’s assessment of the level of previous work that’s been done at the intersection of these 3 areas.  (E.g., the reviewer states our paper “comes in the midst of many other works”; we feel this is erroneous, and revised the paper to make things more clear.)
>
> We have significantly edited the related work section to explain how none of the existing methods directly apply to our setting (e.g., how Triascyn & Faltings 2019 uses a much weaker, empirical measure of privacy than our setting with user-level differential privacy).  In the cases where existing results are applicable, we have in fact used them directly, e.g. adopting techniques from McMahan et al. 2018 and Chen et al. 2019.
>
> Does our revised comparison in the related work section resolve the reviewer’s concerns?

---

### Official Review · AnonReviewer1 · 2019-10-30
**Official Blind Review #1**

**Rating:** 8

**Review:**

Goals
The paper identifies a key challenge in a large class of real world federate learning problems where we also have to ensure user level data privacy. In these settings the modeler can not inspect the raw data samples from the user (due to privacy concerns) and hence all modeling tasks (from data wrangling to hypothesis generation to labeling to model class selection to validation) become far more challenging. The paper proposes that in these circumstances one may use a generative model that learns the data distribution using federated learning methods with provable differentiable privacy guarantees. The generative model can then produce data (unconditional, or conditional on some features or class labels) which can be inspected by the modeler without compromising user privacy.

Experiments
The authors illustrate the approach using existing federated DP RNN learning methods, and using a slightly novel GAN learning algorithm for images (largely similar to other algorithms). They use these methods to provide two examples: 1) learning a language model from text (word sequences) where there is a bug in pre-processing steps (tokenization); 2) learning a GAN for images of handwriting on checks where there is a pre-processing bug that inverts the grayscale of images. These examples illustrate the potential for such methods to possibly be useful to modelers. While one may quibble about some details (see section below) the experimental set up is reasonable to illustrate the need and some of the challenges modelers are likely to face in the real world (

Evaluation & Questions

I'm really torn because I really enjoyed the paper very much overall but I have some strong concerns as well.

Positives: the paper is well motivated and very well written (it is really a pleasure to read, and it is very clear about the details -- especially after they release the code it should be possible to reproduce the results too). The authors shine a spot light on a problem that is very important & widespread (eg while learning from condifential data on cell phones). The proposed solution is fairly simple, intuitive, and quite high level (lets use a generative model that creates phantom data that can be inspected)

Negatives: I am not entirely sold on this being a realistic approach in the long term -- ie that some of the key problems will ever be solvable (I'm quite ok even if they are not solved now in the first paperr). The authors do a very good job of being transparent about several potential issues (see eg last paragraphs of main paper and appendix D). My biggest concerns are below:
* the phantom samples generated from the model need to be very realistic in order to be useful. In other words, we need to have excellent, high fidelity generate models. Even to create proper hypothesis, create proper model classes, assess convergence, or assess whether the generative model is good enough one needs to be able to inspect the raw data -- which can not be done in the first place. This can not be entirely automated eliminating need for human inspection -- and the problem is much worse in generative models (which need to encode more information) than in discriminative models which need to encode less information (bits) almost by definition. Thus one has simply traded the problem of needing to inspect data to model the final algorithm (whcih could be discriminative) and has to deal with the problem of needing data to inspect the intermediate, generative model (which is also learned using federates, DP guaranteeing ways). It is not at all clear what one accomplished by doing this.
* GANs are notoriously hard to train with mode collapse etc.  Setting hyper parameters of any generative model also needs access to original data and impacts the privacy guarantees.

***NOTE added after author response***
The rebuttal has sufficiently address the quibbles I raised below. I'm leaving it here to allow traceability. I'm not fully convinced about the response to the main issue I raised above (ie if the generative model is not very representative, high-fidelity, then one cant know whether a potential bug discerned by inspecting its samples is an artefact of the generative model or whether it is truly a fundamental bug upstream -- and training a high quality generative model also requires one to inspect the raw data in the first place so the problem has simply been swept under the carpet). Nevertheless for a first paper on the topic I think the contributions and intuition provided here are quite valuable so I am ok leaving this for for future work.

* quibble#1: theoretical DP bounds are not very tight. For example, in table 2 they may want to use realistic estimates instead of epsilon even to prove their high level point. I'm not sure I can buy their argument even on this illustrative problem as it stands.
* quibble#2: You may want to at least make an effort to compare against the nearest possible methods in your experimental setup even if they are not a great match to the problem. I'm not intimately familiar with the recent literature but you mention Triascyn & Faltings (2019) so perhaps you could also use that and expand a bit more on the novelty here

**Experience Assessment:**

I have published one or two papers in this area.

**Review Assessment: Checking Correctness Of Derivations And Theory:**

I assessed the sensibility of the derivations and theory.

**Review Assessment: Checking Correctness Of Experiments:**

I assessed the sensibility of the experiments.

**Review Assessment: Thoroughness In Paper Reading:**

I read the paper thoroughly.

---

> ### Author Response · Authors · 2019-11-09
> **Response to Reviewer #1**
>
> We thank the reviewer for their comments and observations, and are thrilled they enjoyed the paper and found the motivating problem and proposed solution compelling.  We now address their list of ‘negatives’ in turn.
>
> Need for Realism:
>
> With regards to the comment that generative models “need to be very realistic in order to be useful”, it is our experience that this is not the case for many real-life problem examples, such as the pixel inversion and concatenation bugs we consider.  The measure of utility for the applications described in our paper is not realism, but rather the ability to detect the presence of distinguishing characteristics in the mimicked distribution.  We agree it is certainly the case that one could not distinguish all characteristics unless generating content to full realism, but it definitely the case that there are a broad set of characteristics that are distinguishable at well short of full realism.  We feel the two problem examples demonstrate this characteristic: while not generating extremely realistic samples, they nevertheless convey a clear ‘signal’ that is useful to the modeler, e.g., the presence of bugs. But our work far from solves the problem we address, and we hope this encourages new work in the generative modeling community.
>
> Thanks for the reviewer’s comment as we’ve updated the paper (e.g., Section 2.1) to better describe the types of problems where lower-fidelity synthesis is ok and the problems where high-fidelity will likely be necessary. Please let us know if this addresses the reviewer’s concerns.
>
> Privacy budgets for hyperparameter sweeps (“Setting hyper parameters of any generative model also needs access to original data and impacts the privacy guarantees. ”):
>
> We believe what you are referring to is that identifying the correct hyperparameters typically requires a ‘sweep’ of values, each of which involves data queries against the private data; the privacy budget must account for all these queries, not simply the final training run.  (If we have mistakenly interpreted your comment, we apologize, and would benefit from a clarification.)
>
> This is absolutely true, and we made sure this paper raises this issue prominently.  In Section 3 on DP Federated Generative Models, we conclude the discussion of DP by noting “....  that since the modeler has access to not only the auxiliary generative models, but also potentially other models trained on the same data, if an (eps, delta) guarantee is desired, the total privacy loss should be quantified across these models, e.g. ...”  We also discuss the need for algorithms requiring minimal tuning as an import step for future work. Again, our contribution is primarily in highlighting this important problem, rather than solving it. Please let us know if you feel our current wording doesn’t properly convey this matter prominently enough; we certainly wish to call attention to it as we hope to see further research in this area.
>
> It’s also true that this is a broader concern that impacts not just the generative models of our paper, but any ML (or other query-based) process that repeatedly samples from private data.  There are some mitigations typically proposed (i.e., using a different, proxy dataset to work out the hyperparameter values before training on the actual private data), but this continues to be an active research area in the larger DP community, which we applaud.  Along with benefiting everyone else working in DP ML (generative models or not, federated or not), it will definitely benefit those of us working with federated generative models.
>
> Finally, as the paper shows, GAN convergence did not take an exorbitant # of rounds (the generated image results we show are after 1000 rounds of federated training).  So the volume of data queries being performed when training federated generative models is in-line with the typical volume of data queries performed when doing any type of federated learning.
>
> Quibble #1 - DP bounds:
>
> Would the reviewer be able to clarify this comment further for our benefit?  We regretfully have had trouble parsing their meaning the first time around.  As DP gives us an upper bound on privacy loss, and we’re achieving a DP $(\epsilon, \delta)$ at population scale that are indicative of a tight bound, we feel we’ve shown that privacy loss is minimal?  We must be misunderstanding something in the reviewer’s comment/critique.
>
> Quibble #2 - Compare with other methods:
>
> We have significantly edited the related work section to explain how none of the existing methods directly apply to our setting; in the cases where existing results are applicable, we have in fact used them directly, e.g. adopting techniques from McMahan et al. (2018) and Chen et al. (2019). Please let us know if this does not address these concerns.

---

### Decision · Program_Chairs · 2019-12-19

**Decision:**

Accept (Poster)

**Comment:**

The paper provides methods for training generative models by combining federated learning techniques with differentiable privacy. The paper also provides two concrete applications for the problem of debugging models. Even though the method in the paper seems to be a standard combination of DP deep learning and federated learning, the paper is well-written and presents interesting use cases.